# Efficient Photodegradation of Rhodamine B by Fiber-like Nitrogen-Doped TiO_2_/Ni(OH)_2_ Nanocomposite under Visible Light Irradiation

**DOI:** 10.3390/mi14040870

**Published:** 2023-04-18

**Authors:** Huan Wang, Mingxuan Dong, Baorui Shao, Yaodan Chi, Chao Wang, Sa Lv, Ran Duan, Boqi Wu, Xiaotian Yang

**Affiliations:** 1Key Laboratory for Comprehensive Energy Saving of Cold Regions Architecture of Ministry of Education, Jilin Jianzhu University, Changchun 130118, China; wanghuan@jlju.edu.cn (H.W.); wangchao@jlju.edu.cn (C.W.); lvsa82@163.com (S.L.); 18186880816@163.com (R.D.); xiancaitang@sina.com (B.W.); 2Department of Materials Science, Jilin Jianzhu University, Changchun 130118, China; dong990816@163.com (M.D.); br13180828503@163.com (B.S.); 3Department of Chemistry, Jilin Normal University, Siping 136000, China

**Keywords:** photocatalysis, charge transfer, heterostructure

## Abstract

N-TiO_2_/Ni(OH)_2_ nanofiber was successfully prepared by combining the electrospinning and solvothermal method. It has been found that under visible light irradiation, the as-obtained nanofiber exhibits excellent activity for the photodegradation of rhodamine B, and the average degradation rate reaches 3.1%/min^−1^. Further insight investigations reveal that such a high activity was mainly due to the heterostructure-induced increase in the charge transfer rate and separation efficiency.

## 1. Introduction

Back in 2001, Asahi et al. [1,2] identified that the tactic of non-metal-doping can narrow the bandgap of TiO_2_ (anatase, 3.2 eV). Both experimental results and the relevant density functional theory (DFT) theoretical calculations revealed that nitrogen doping could significantly extend the absorption range of TiO_2_ into the visible region. Since then, numerous investigations focus on non-metal-doped (N, S, P, B) TiO_2_ photocatalysts [3,4]. Through the non-metallic dopants and their combinations, the corresponding photocatalytic activity of derived TiO_2_ could be improved, thus exhibiting outstanding photocatalytic performances in various applications, such as photocatalytic water splitting [5,6], CO_2_ reduction [7,8], removal of organic compounds [9,10], sterilization [11], and anti-viral applications [12]. Owing to these excellent properties, N-doped TiO_2_ and cocatalyst-supported TiO_2_ have been commercialized and widely used in many practical applications [13]. However, the photocatalytic reaction rate of N-doped TiO_2_ is still very low, which is believed to be caused by the following two reasons: one is its weak absorption in the visible range, while the other is the limited rate of the photocatalytic reactions by the nitrogen-induced oxygen vacancies, which is supposed to act as the carrier traps and/or recombination centers.

To enhance the photocatalytic performance of nitrogen-doped TiO_2_ (N-TiO_2_) in the visible range, tremendous efforts have been devoted to it during the past decades [14,15,16]. It has been revealed that transition metal nickel hydroxide and NiS could act as cocatalysts to significantly strengthen the photocatalytic activity of TiO_2_, which is believed to be attributed to the inhibition of charge carriers’ recombination [17,18,19]. Zhang et al. [20] prepared three-dimensional flower-like TiO_2_@Ni(OH)_2_ core–shell heterostructures, exhibiting a six-fold activity enhancement for hydrogen photogeneration. Meng et al. [21] synthesized a hierarchical TiO_2_/Ni(OH)_2_ composite photocatalyst via decorating Ni(OH)_2_ nanosheets on the electrospun TiO_2_ nanofibers by a simple wet method, and the results indicate that, compared to pristine TiO_2_ fibers, its activity of CO_2_ photoreduction was significantly improved, which was ascribed to the increased charge separation efficiency and higher CO_2_ capture capacity owing to the presence of Ni(OH)_2_. Chen et al. [22] reported the construction of a photo-supercapacitor with high specific capacitance and ultra-fast charge–discharge response, in which TiO_2_ was used as a photoelectric conversion material and Ni(OH)_2_ was used as an energy storage material in the TiO_2_/Ni(OH)_2_ composite material, offering a simple fabrication process for efficient direct solar energy storage. These findings inspired us to develop a heterojunction between N-TiO_2_ and nickel hydroxide to increase the charge separation efficiency thereby improving its corresponding photocatalytic activity.

To our knowledge, only a few studies were focused on the photocatalytic applications of N-TiO_2_/Ni(OH)_2_ composite. In this work, we first prepared N-TiO_2_ nanofibers by electrospinning technique followed by decoration of Ni(OH)_2_ via the hydrothermal method to obtain the high-quality N-TiO_2_/Ni(OH)_2_ heterostructures. The designated composite exhibits a much-enhanced photocatalytic ability for the degradation of rhodamine B (RhB) compared to N-TiO_2_ under visible light irradiation. Further detailed analyses reveal the relevant enhanced activity is ascribed to the increased charge separation efficiency induced by the delicate heterostructure.

## 2. Experimental Section

### 2.1. Chemicals and Reagents

Acetic acid (99.5%), butyl titanate (99.0%), ethanol (99.0%), urea (99.0%), polyvinylpyrrolidone (99.0%), Rhodamine B (99.0%), nickel nitrate hexahydrate (99.0%), and hexamethylenetetramine (99.0%) were purchased from Shanghai Aladdin Biochemical Technology Co., Ltd. (Shanghai, China). All chemicals were used as received without further purification.

### 2.2. Synthesis of Ni(OH)_2_ Nanosheets

N-TiO_2_ nanofibers were prepared by the electrospinning method according to the previous work [23]. Typically, 5 mL of acetic acid and 5 mL of butyl titanate were added into 10 mL ethanol under vigorous stirring at room temperature (RT). After ~10 min., 0.2 g of urea was introduced, followed by another 20 min stirring. Meanwhile, another solution was obtained by introducing 1.5 g of polyvinylpyrrolidone into 10 mL ethanol. Then, the two above solutions were homogeneously mixed and further heated at 70 °C for 1 h under vigorous stirring to obtain the precursor solution for electrospinning. At a static voltage of 15 kV, a capillary tube with an inner diameter of 0.5 mm at a 15 cm distance from the collector was used to prepare the electrospun film. The flow rate was set at 1 mL/h. Afterward, the fiber film was peeled off and placed in a crucible that was sealed with tin foil and kept at 550 °C for 3 h in a muffle furnace to obtain N-TiO_2_ fibers. For comparison, TiO_2_ nanofibers were prepared by a similar procedure and the only exception is without introducing urea.

### 2.3. Preparation of N-TiO_2_/Ni(OH)_2_ Nanocomposite

A total of 100 mg of the as-prepared N-TiO_2_ nanofibers were added into 40 mL ultrapure water followed by sonication for 10 min to obtain a homogeneous suspension. Next, 3.1 mg nickel nitrate hexahydrate and 1.7 mg hexamethylenetetramine were introduced and the corresponding solution was further stirred for 1 h at RT. Then, the solution was transferred to the Teflon-lined autoclave and heated at 120 °C for another 6 h. The obtained precipitation was alternatively washed with ethanol and water several times and subsequently dried in a vacuum oven at 80 °C for 4 h. Samples with 0.5 and 1.5 wt% Ni(OH)_2_ contents were prepared by the same protocol with different amounts of nickel nitrate hexahydrate and hexamethylenetetramine introduced.

### 2.4. Characterization

The crystal structure of samples was investigated on an X-ray diffractometer (Rigaku D/Max 2550, Japan Rigaku Co., Ltd., Tokyo, Japan) at 40 kV and 40 mA with copper Kα radiation (λ = 0.154056 nm). The morphological and structural information was characterized via field-emission scanning electron microscopy (SEM, JSM-7610F, Tokyo, Japan) and transmission electron microscopy (TEM, Tecnai G2 F20 S-TWIN, FEI, Valley City, ND, USA). X-ray photoelectron spectroscopy (XPS) measurements were recorded by a photoelectron spectrometer (ESCALAB MKII, VG Scientific, Waltham, MA, USA). The relevant photoluminescence results were obtained on an FLS-1000 fluorescence spectrophotometer (Edinburgh Instruments Ltd., Livingston, UK). The transient photocurrent signals were measured by an electrochemical workstation (CHI-660, CH Instruments, Austin, TX, USA). UV-Vis diffuse reflectance spectra of the samples, and the absorbance of Rh B solution was measured on a Shimadzu-2600 spectrophotometer (Shimadzu, Kyoto, Japan).

### 2.5. Measurement of Photocatalytic Reactions

The RhB photodegradation experiments were performed in a glass flask. Typically, 25 mg of the relevant photocatalyst was dispersed into a 100 mL Rh B aqueous solution (10 mg/L). Before the photoreaction, the solution was stirred for 30 min in dark to exclude the adsorption effect of the sample. Then, the solution was illuminated by a 300-W Xenon lamp with a filter (>420 nm). A total of 2 mL of the solution was taken at regular intervals (30 min), then subjected to UV-Vis measurements after filtrating the photocatalyst.

## 3. Results

The crystal structural information of TiO_2_, N-TiO_2_, and N-TiO_2_/Ni(OH)_2_ nanofibers were obtained by X-ray diffraction (XRD) technique. As shown (Figure 1), for TiO_2_, the clear diffraction peaks at 2θ = 25.2°, 37.8°, 47.8°, 55.2°, 62.7°, 68.9°, 75.1°, and 82.6° ascribed to (101), (004), (200), (105), (204), (116), and (215) crystal planes of anatase (JCPDS card number 21-1272) were observed, respectively. Meanwhile, peaks at 2θ = 27.4°, 36.1°, and 41.1°, which is consistent with the (110), (101), and (111) crystal planes of rutile, respectively, TiO_2_ (JCPDS card number 21-1276) are also detected. The results demonstrate that under the current calcination condition, the obtained TiO_2_ nanofibers are in the mixed phase of anatase and rutile. Interestingly, when nitrogen is introduced into TiO_2_, the rutile phase of TiO_2_ seems to be significantly inhibited. Furthermore, for N-TiO_2_/Ni(OH)_2_ nanocomposite, no peak belonging to Ni(OH)_2_ was observed, which is considered to be owing to the low Ni(OH)_2_ content [18].

The morphological and structural information of the sample was obtained by SEM and TEM measurements. Figure 2a displays the SEM result of TiO_2_ nanofibers. It is seen that the TiO_2_ was in a fibrous structure with a diameter of ~500 nm and length over 10 µm, while the introduction of nitrogen did not induce any observable morphological change (Figure 2b). Further decoration of Ni(OH)_2_ onto N-TiO_2_ nanofibers could induce the two-dimensional structural growth on the surface of the corresponding samples, (Figure 2c), which manifests the successful deposition of Ni(OH)_2_. TEM image of the N-TiO_2_/1.0 wt% Ni(OH)_2_ nanofibers indicates the diameter of the relevant sample in ~500 nm (Figure 2d), and higher magnification TEM result clearly demonstrates a 50-nm-thick layer was homogenously coated onto the outside surface of the N-TiO_2_ fibers, which is considered to be Ni(OH)_2_ (Figure 2e). Furthermore, HRTEM measurement confirms our assumption. As is shown (Figure 2f), the clear granular nanostructure ascribed to (110) crystal plane of Ni(OH)_2_ was found in the corresponding outside layer area of the sample, and the green circle represents Ni(OH)_2_ nanocrystalline particles in Figure 2f.

To acquire the chemical state of each element in the samples, XPS measurements were performed. Figure 3 shows the high-resolution Ti 2p, O 1s, N 1s, and Ni 2p XPS spectra of the N-TiO_2_/1.0 wt% Ni(OH)_2_. As is shown for Ti 2p, peaks at 458.5 and 464.3 eV could be ascribed to Ti 2p_3/2_ and Ti 2p_1/2_ of TiO_2_, respectively, indicating the existence of Ti^4+^ in the sample [24,25,26]; while for O 1s, signals of lattice oxygen (titanium–oxygen–titanium, 529.7 eV) and hydroxyl oxygen (Ti-OH and Ni-OH, 531.4 eV) are observed [27] (Figure 3b). A weak characteristic peak (N1s) at 399.9 eV attributes to the signal combination of weakly charged nitrogen species with C, H, or O atoms [28] (Figure 3c), and demonstrates the successful diffusion of nitrogen into the TiO_2_ lattice at interstitial positions [29,30]. It is noted that nitrogen had a weak positive charge in the Ni 2p region (Figure 3d) due to the bonding with O atoms of TiO_2_ [30,31,32]. In addition, the peaks at 855.3 and 873.0 eV correspond to Ni 2p_3/2_ and Ni 2p_1/2_, respectively, indicating the existence of nickel hydroxide [33]. In addition, the two shoulder peaks were ascribed to the satellite peaks of Ni 2p_3/2_ and Ni 2p_1/2_ in the vicinity of 861.7 and 880.3 eV, respectively [20].

The absorption property of different samples was measured by UV-Vis diffuse reflectance absorption spectroscopy. As indicated (Figure 4), Ni(OH)_2_ control sample exhibits an obvious absorption at 450 and 600–800 nm, which is due to the d–d transition of Ni [34]. In addition, compared to TiO_2_, the absorption of N-TiO_2_ and different N-TiO_2_/Ni(OH)_2_ samples in the visible region are significantly enhanced. It should be noted that, in the range of 400–550 nm, the absorption intensity of N-TiO_2_/Ni(OH)_2_ nanofibers is increased with the increase in Ni(OH)_2_ content.

The photocatalytic activity on the degradation of RhB of different samples was then evaluated. As shown (Figure 5a), all the employed samples exhibit the similar physical adsorption property for RhB. However, interestingly, once illuminated, all the N-TiO_2_/Ni(OH)_2_ nanocomposites show a much higher ability for the photodegradation of RhB compared to that of a relevant single component. Especially, N-TiO_2_/Ni(OH)_2_ nanocomposite possesses the best performance, and ~97% of RhB would be photodegraded in 90 min. To illustrate the decomposition rate more clearly, the Langmuir-Hinshelwood kinetics equation was employed [35]. As shown in Figure 5b, the activities of all the nanocomposites were higher than those of a single component, indicating an apparent synergistic effect between N-TiO_2_ and Ni(OH)_2_, which could also be reflected by the rate constant (Figure 5c). Our results unambiguously manifest the vital role of forming the heterostructure for significantly enhancing the RhB photodegradation ability. Further optimization of the addition of Ni(OH)_2_ in the composite showed that when the amount of Ni(OH)_2_ in the composite is 1.0-wt%, the best performance of 0.0310 min^−1^ could be achieved (Figure 5b), which was comparable to the recent benchmark test results (Table 1).

The stability of photocatalyst is a critical issue for the potential real application. Hence, the relevant stability of N-TiO_2_/1.0-wt% Ni(OH)_2_ was evaluated (Figure 5d). After each round of photocatalytic reaction, the sample was collected via centrifugation and washed with ultrapure water several times before further usage. As shown, only a slight activity decrease (~3.7%) was observed after five cycles demonstrating its excellent stability for long-term use.

To elucidate the underlying mechanism of the heterostructure-induced photocatalytic activity enhancement, photoluminescence (PL) measurements were first employed. As shown in Figure 6a, N-TiO_2_ gives an obvious PL emission in the range of 400–550 nm, while for that of Ni(OH)_2_, hardly any PL signal could be observed. Interestingly, for the nanocomposite sample, the relevant PL intensity is gradually decreased with the gradual increase in Ni(OH)_2_ content, which is considered to be ascribed to charge transfer from N-TiO_2_ to Ni(OH)_2_ [35]. To further verify the separation and migration behavior of charge carriers, photocurrent response tests were performed. As shown in Figure 6b, all the composite samples give higher photocurrent responses than that of the N-TiO_2_ or Ni(OH)_2_, and N-TiO_2_/1.0 wt% Ni(OH)_2_ exhibits the highest response, which is consistent with the trend of their photocatalytic performance. This strongly signifies the introduction of Ni(OH)_2_ is beneficial for increasing charge separation efficiency, and thus enhances the relevant photocatalytic activity.

Based on the above results, the plausible underlying mechanism for the enhanced photocatalytic activity of N-TiO_2_/Ni(OH)_2_ composite has been proposed (Figure 7). After the absorption of the incident light, both N-TiO_2_ and Ni(OH)_2_ were excited. Thereafter, the electrons and holes are photogenerated in each component, and electrons are excited into the conduction band (CB) while the holes are left at the valence band (VB). Owing to the potential of Ni^2+^/Ni (0.23 eV) being slightly lower than that of anatase TiO_2_ (~0.26 eV), photogenerated electrons will transfer from CB of TiO_2_ into CB of Ni(OH)_2_, while holes remain in VB of TiO_2_. This will significantly promote charge separation and inhibit the recombination of photogenerated electrons and holes. Subsequently, electrons could react with adsorbed oxygen to generate superoxide free radicals. It is noted the formation of free radicals can not only effectively inhibit electron-hole recombination, but also destroy the bonds of RhB and therefore degrade them, which finally exhibits enhanced photocatalytic activity.

## 4. Conclusions

In this work, the heterostructured N-TiO_2_/Ni(OH)_2_ nanofibers were successfully constructed and further applied for the enhanced photodegradation of RhB compared to that of N-TiO_2_ under visible light irradiation (>420 nm). Our results indicate that the elaborated design could significantly enhance the corresponding photocatalytic activity and the one with 1.0 wt% Ni(OH)_2_ exhibits the best performance. Further detailed investigations revealed that the enhanced activity is mainly ascribed to the efficient charge transfer and separation in N-TiO_2_ induced by the integration of Ni(OH)_2_. It is anticipated that our tactics and results would be beneficial for the future design of high-performance photocatalysts.

## Figures and Tables

**Figure 1 micromachines-14-00870-f001:**
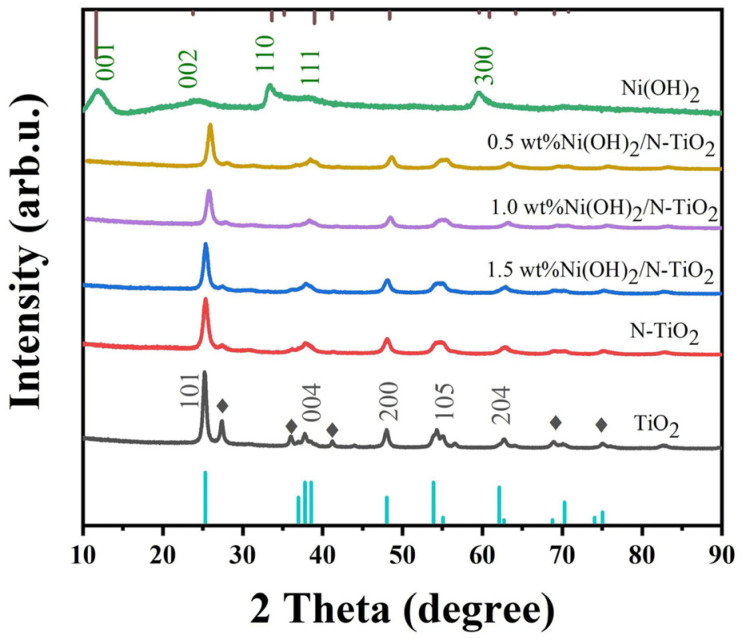
X-ray diffraction (XRD) patterns of TiO_2_, N-TiO_2_, Ni(OH)_2_, and N-TiO_2_/Ni(OH)_2_ with different amount of Ni(OH)_2_. The rectangle indicates the rutile phase of TiO_2_.

**Figure 2 micromachines-14-00870-f002:**
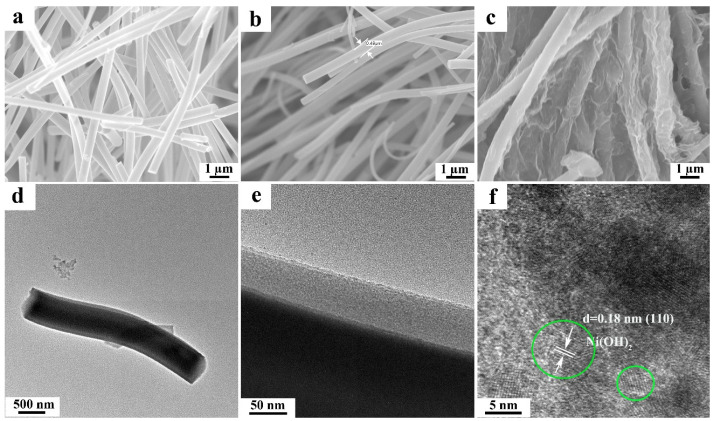
SEM, TEM, and high-resolution TEM (HRTEM) images of (**a**) TiO_2_ fibers, (**b**) N-TiO_2_ fibers, and (**c**–**f**) N-TiO_2_/1.0 wt% Ni(OH)_2_ composite fibers.

**Figure 3 micromachines-14-00870-f003:**
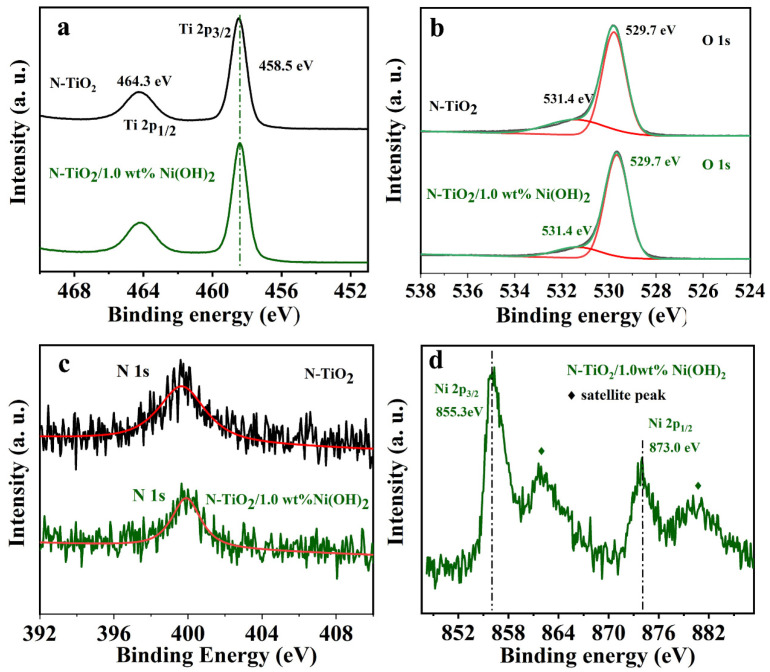
XPS spectra of (**a**) Ti 2p, (**b**) O 1s, (**c**) N 1s, and (**d**) Ni 2p of N-TiO_2_/1.0 wt% Ni(OH)_2_ nanofibers. The red lines in (**b**,**c**) are the fitted curves of the corresponding data.

**Figure 4 micromachines-14-00870-f004:**
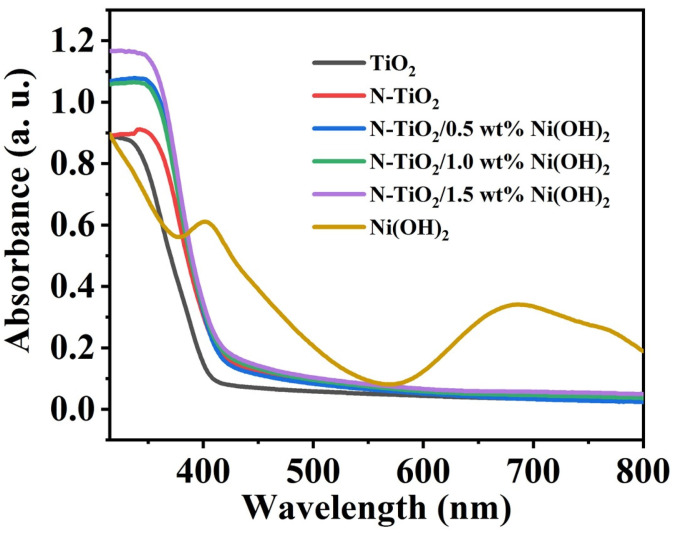
UV-Vis diffuse reflectance absorption spectra of Ni(OH)_2_, TiO_2_, N-TiO_2_, N-TiO_2_/Ni(OH)_2_ with different Ni(OH)_2_ contents, respectively.

**Figure 5 micromachines-14-00870-f005:**
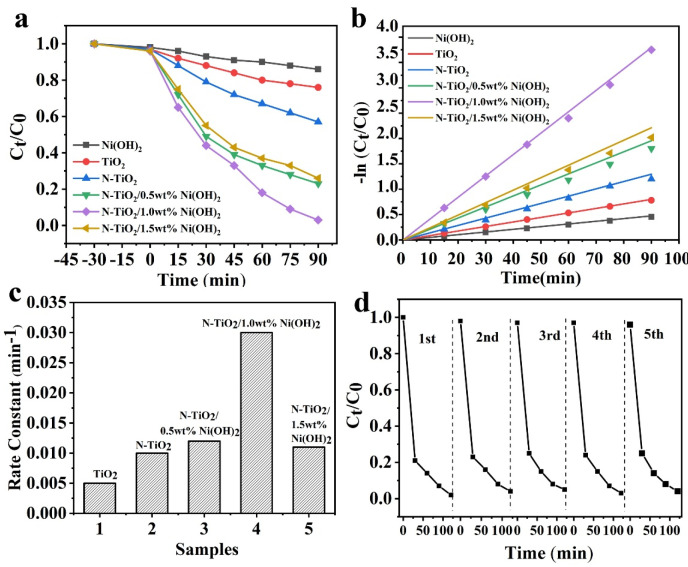
(**a**) Photodegradation of RhB for TiO_2_, N-TiO_2_, Ni(OH)_2_, and N-TiO_2_/Ni(OH)_2_ of different Ni(OH)_2_ contents under the irradiation of visible-light (>420 nm); (**b**) Natural logarithm degradation curves of different samples; (**c**) Rate constants (kR) of different samples for photodegradation of RhB; (**d**) five consecutive photodegeration curves of RhB over N-TiO_2_/1.0-wt% Ni(OH)_2_ under the visible light irradiation.

**Figure 6 micromachines-14-00870-f006:**
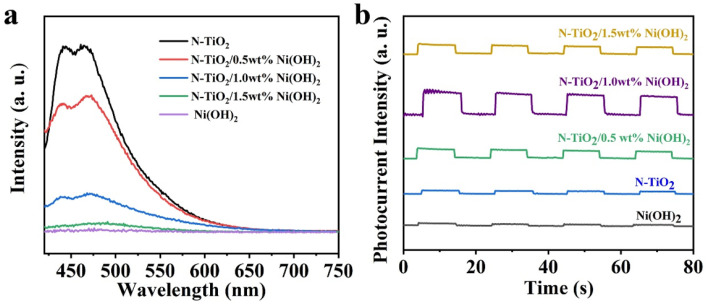
(**a**) PL spectra and (**b**) photocurrent response (>420 nm)of N-TiO_2_, Ni(OH)_2_, and N-TiO_2_/Ni(OH)_2_ with different Ni(OH)_2_ content, respectively.

**Figure 7 micromachines-14-00870-f007:**
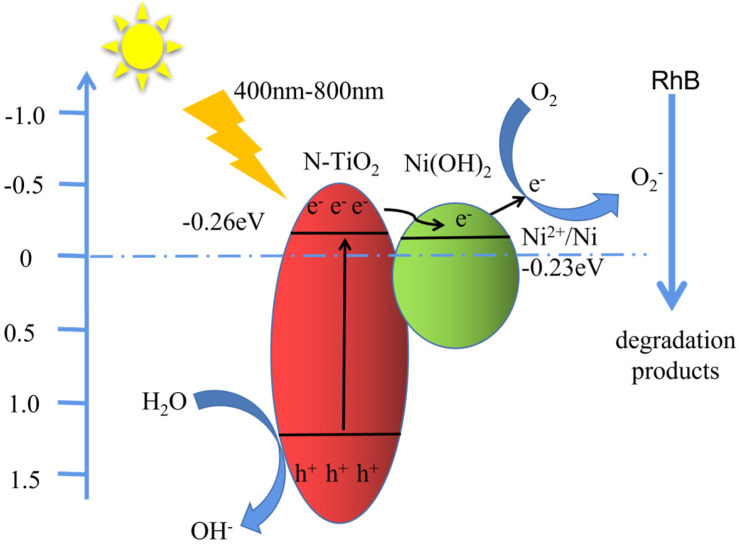
Schematic diagram of the mechanism of photocatalytic degradation of Rh B by Ni(OH)_2_/N-TiO_2_ under visible light irradiation.

**Table 1 micromachines-14-00870-t001:** Comparison of the catalytic performance of N-TiO_2_/Ni(OH)_2_ with the previously reported data for the reduction in organic dyes of Rh B.

Material Used	Light Used	Dye Degradation	K_app_ (min^−1^)	Removal (%)/ 30 min	Ref.
N-TiO_2_/Ni(OH)_2_	visible light	RhB	0.0310	60%	This work
Ni(OH)_2_	UV-visible	RhB	0.0130	/	[36]
NiO-ZnO	UV-visible	RhB	0.0302	50	[37]
NixOy/TiO_2_	UV-visible	RhB	0.0349	56.5	[38]
Ni/NiO/TiO_2_	UV-visible	RhB	/	35	[39]
NiO/BVO	UV-visible	RhB	/	40	[40]

## Data Availability

The data presented in this study are available on request from the corresponding author.

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
