# Peer review of "Efficient Photodegradation of Rhodamine B by Fiber-like Nitrogen-Doped TiO2/Ni(OH)2 Nanocomposite under Visible Light Irradiation"

_micromachines, 2023, doi:10.3390/mi14040870_

Round 1

Reviewer 1 Report

Comments to the Author

  The article by Wang and co-workers reports on the synthesis and photocatalytic activities of the N-TiO2/Ni (OH)2 nanofibers, in which the as-obtained nanofibers exhibit an excellent activity for the photodegradation of rhodamine B under visible light irradiation. I suggest the authors to revise the paper carefully to make it more readable.

   Here are some concerns that prevent the publication of the manuscript.

(1) How about the stability of the N-TiO2/Ni (OH)2 during photocatalytic degradation? Whether it can be reused?

(2) To effectively reflect the advantages of N-TiO2/Ni (OH)2, please discuss its photocatalysis mechanism of Ni (OH)2 -enhanced photocatalytic activity.

(3) The author mentioned that the average degradation rate of N-TiO2/Ni(OH)2 photocatalysts reached 3.1%/min-1. What are the advantages compared to other studies.

(4) They should further improve the English writing of whole paper. The mistake should be revised, such as “photodegeration” in the conclusion section.

Reviewer 2 Report

In this paper, entitled “Efficient Photodegradation of Rhodamine B by Fiber-like Nitrogen-doped TiO2/Ni(OH)2 Nanocomposite under Visible Light Irradiation”, Yang et al. prepared fiber-like N-doped TiO2/Ni(OH)2 nanocomposite by combining the electrospinning and solvothermal method. Then compared the photodegradation of rhodamine B with TiO2, N-TiO2, Ni(OH)2, and N-TiO2/Ni(OH)2 with different Ni(OH)2 contents. The results showed that the elaborated heterostructure design enhanced the photocatalytic activity, and the one with 1.0 wt% Ni(OH)2 exhibited the best performance. In my opinion, the results reported in this manuscript are worthy of attention. Therefore, I suggest publishing it in Micromachines after minor revisions.

1. Experimental part on page 2, the supplies of materials and their degree of purity must be added. Additionally, the standard XRD peaks shown in Figure 1 should be made more distinguishable.

2. On page 6, the authors claimed that “the relevant PL intensity is gradually decreased with the gradual increase of Ni(OH)2 content, which is considered to be ascribed to charge transfer from N-TiO2 to Ni(OH)2”. I am curious if the decrease in PL intensity can be attributed to the addition of Ni(OH)2. This is because the neat Ni(OH)2 shows almost no peak, whereas the peak gradually increases upon adding the N-TiO2. The authors should provide more experimental evidence to support this claim.

3. How about the reusability of the photocatalyst? Can the photocatalyst be regenerated?

4. Comparison of the photocatalytic performance of N-TiO2/Ni(OH)2 composites with the other reported works would emphasize the claimed advantages of this study.

5. The language should be polished. For example, “much effort has been devoted to improve the photocatalytic activity of nitrogen-doped TiO2 (N-TiO2) in visible range” on page 1, line 41; “N-TiO2 nanofibers was prepared by electrospinning technique” on page 2, line 69; “matches well with the trend of the their photocatalytic performance” on page 6, line 198; et al.
